# Predictors of neonatal mortality among neonates in Tigray regional state, Ethiopia: A cross-sectional study

**Gebru Gebremeskel Gebrerufael**[1]*, **Brhane Gebrehiwot Welegebrial**[2], **Mehari Gebre Teklezgi**[1]

1 Department of Statistics, College of Natural Science, Adigrat University, Adigrat, Ethiopia, 2 Department of Pharmacy, College of Medicine and Health Sciences, Adigrat University, Adigrat, Ethiopia

* gebrugebremeskel2015@gmail.com

## Abstract

### Background

Since 2015, Ethiopia is committed to lowering the death rate for children under five and it is one of the countries in Sub-Saharan Africa that has accomplished the fourth Millennium Development Goal. However, in Ethiopia, neonatal death has remained a serious public health concern, with greater rates found in Tigray regional state and the predictors aren't well recognized. The goal of this study was to ascertain the prevalence of neonatal death in the Tigray regional State as well as any relevant risk factors.

### Methods

This study performed a secondary data analysis of the 2016 Ethiopia Demographic and Health Survey (EDHS) report. Information was gathered on 716 neonates who were born five years before the survey began. Risk factors for neonatal mortality were thought to include mother and neonate demographics, health, and environmental factors. The study employed multivariable logistic regression model analysis and descriptive statistics to identify significant correlates of neonatal mortality.

### Results

In Tigray regional state, the overall prevalence of neonatal mortality was 4.3% (95% CI: 3.06, 6.10). The multivariable logistic regression model analysis revealed that factors such as multiple birth types (AOR = 15.3, 95% CI: 3.54, 65.84), birth order (2–4) (AOR = 4.88, 95% CI: 1.52, 15.7), sex of the neonate (being male) (AOR = 3.75, 95% CI: 1.45, 9.75), home place of delivery (AOR = 7.4, 95% CI: 2.0, 27.6), and neonates born to mothers aged 20–34 years (AOR = 0.23, 95% CI: 0.087, 0.58) were significantly risk factors associated with a higher risk of neonatal mortality rate.

**Data Availability Statement:** Datasets cannot be shared publicly because they contain sensitive participant information. Datasets are available from the EDHS after approval from the IRB unit. Access to the study data set requires legal registration at

https://dhsprogram.com/data/available-datasets. cfm and the creation of a persuasive letter outlining the project descriptive goal for the DHS program. More access information can also be found on the DHS Program website (https://dhsprogram.com/ data/Access-Instructions.cfm). The authors confirm that interested researchers would be able to access these data in the same manner as the authors. The authors also confirm that they had no special access privileges that others would not have.

**Funding:** The author(s) received no specific funding for this work.

**Competing interests:** The authors have declared that no competing interests exist.

**Abbreviations:** ANC, **A**ntenatal **C**are; AOR, **A**djusted **O**dds **R**atio; CI, **C**onfidence **I**nterval; CSA, **C**entral **S**tatistical **A**gency; EDHS, **E**thiopian **D**emographic **H**ealth **S**urvey; EPHI, **E**thiopian **P**ublic **H**ealth **I**nstitute; LRT, **L**ikelihood **R**atio **T**est; MDG4, **M**illennium **D**evelopment **G**oal; MOH, **M**inistry of **H**ealth; NMR, **N**eonatal **M**ortality **R**ate; SPSS, **S**tatistical **P**ackage of **S**ocial **S**cience; SSA, **S**ub-**S**aharan **A**frica; UNICEF, **U**nited **N**ations **I**nternational **C**hildren's **E**mergency **F**und; USAID, **U**nited **S**tates **A**gency for **I**nternational **D**evelopment.

## Conclusions

The study recognized the sex of the neonate, birth order, mother's age, place of delivery, and birth type as potential risk factors for neonatal mortality. The prevalence of neonatal mortality indicated that the neonatal mortality rate in Tigray regional state was higher than the national average. To reduce neonatal mortality, targeted interventions should focus on high-risk groups, such as mothers delivering at home and those with multiple births.

## Background

Neonatal mortality rate (NMR) refers to the probability of a neonate dying within the first 28 days of life [1]. Neonatal care lasts for 28 complete days after delivery, starting at birth. At this point, a newborn's chances of survival are at their lowest [2]. Every year, 2.9 million newborns worldwide pass away in their first month of life; in 2014, the majority of these deaths happened in developing countries [3]. In developing countries, the survival rate of preterm infants is extremely low [3]. Neonatal mortality accounts for approximately 44% of deaths in children under five, and impoverished countries account for more than 99% of these deaths [1, 4, 5]. Global public health continues to be greatly concerned about neonatal mortality [2–4].

Furthermore, Sub-Saharan Africa (SSA), which has the highest rate of neonatal mortality worldwide and the least success in lowering infant mortality, is where this trend is most pronounced [6].

Moreover, gaining insight into the root causes of infant deaths is essential to making substantial strides in public health and reducing the rates of neonatal and maternal mortality. This can help as, policymakers and designers have developed suggestions and addressed these linked predictor factors with appropriate interventions in order to enhance the public health status of mothers and newborns. The creation and preparation of pertinent guidelines and policies will safeguard the sustainability of achieving a decrease in the early infant mortality rate [7]. Ethiopia is ranked sixth internationally and second in the SSA after Nigeria, making it one of the countries with the highest NMR contributions [8].

Among the most significant predictors of neonatal mortality are the sex of the neonate [9], antenatal care (ANC) follow-up [10, 11], birth weight (low birth weight) [10, 11], preterm birth, fetal growth restriction, and congenital abnormalities [12]. Therefore, to lower a child's high NMR, the incidence of issues during the initial neonatal period, and to ensure the survival of neonatal babies, professional delivery support, excellent antenatal care, and postnatal care follow-up should be implemented.

Over the 15 years prior, there has been a steady decline in Ethiopia's baby and under-five death rates. However since 2011, NMR has raised from 29 live births per 1000 to 37 live births per 1000. As a result, this report shows that it has still continued to indicate at a high level. The 2016 Ethiopia Demographic and Health Surveys (EDHS) [13, 14] show that the NMR of 34 fatalities per 1000 births in the Tigray National Regional State is higher than the national average of 30 deaths per 1000 pregnancies.

As of 2015, Ethiopia was among the SSA countries that had achieved the fourth Millennium Development Goal (MDG4), which aims to lower the mortality rate of children under five. Despite various studies on neonatal mortality in Ethiopia, research focused specifically on regional variations, especially in the Tigray region, is limited. Thus, in order to continue working toward this goal in the future, a considerable drop in NMR is necessary. Previous research indicates that during the last thirty years, early NMR has declined more slowly than late NMR.

Interventions to lower neonatal mortality are a key concern and are part of improving maternal public health care because it is vital to oversee the current health programs and set policies on improving the current situation for mother and neonate health status. Furthermore, knowledge of the risk factors associated with neonatal mortality is necessary to monitor the effectiveness of intensive and evidence-based public health programs to reduce neonatal fatalities [15, 16].

Ethiopia has seen a large number of studies looking into potential risk factors for baby and under-five child mortality. Studies that look at the NMR and the associated risk factors are, however, scarce. Most of them are concentrated at the federal level. Because the results by country level may not accurately reflect the situation at the regional level, planners and policy-makers should take this into mind, something that these studies neglect to do.

To address this identified gap, we carried out a thorough retrospective cross-sectional study design analysis of the most recent EDHS 2016 to determine the main risk factors associated with NMR in the Tigray regional state. We considered a variety of socioeconomic, demographic, health, and environmental predictor factors in the Tigray Regional State, Ethiopia [13]. Therefore, the objective of this study was to determine the prevalence and predictors of neonatal mortality in the Tigray region of northern Ethiopia.

## Method and materials

### Study area

Tigray regional state is located between 36-and 40-degrees east longitude north of Ethiopia, bordering Eritrea to the north, Amhara to the south, Sudan to the west, and Afar to the east. 3,136,267 populations, of whom 1,542,165 were men, and 2,667,789 (85%) of them lived in rural regions were found in the Tigray region state, according to the 2007 Ethiopian population and housing censuses [17].

### Study design, source of data, and period

Based on the EDHS 2016 dataset, this study employed a cross-sectional study design with institutionally based secondary data analysis. The survey was conducted in January and June of 2016 by the Ministry of Health (MOH), the Central Statistics Agency, and the Ethiopian Public Health Institute (EPHI). Sponsorship was provided by the United States Agency for International Development (USAID). The major objective of this survey report is to give policymakers and designers detailed information on fertility, adult, newborn, child, and maternal mortality, as well as the health status of mothers and children, nutrition, and awareness of HIV/AIDS and other sexually transmitted illnesses (STDs).

### Sample size determination technique and study population

This 2016 EDHS survey report employed a two-step sample selection study design. Ethiopia is divided into two administrative cities and nine regional states. Using a probability proportionate to size, 645 enumeration areas, 443 in rural and 202 in urban areas, were selected for the survey's first phase. The newly created household list was methodically chosen in the second stage, with 28 homes per cluster being chosen with an equal probability. The interview was open to all reproductive women aged 15 to 49 who were stable members of the selected homes and who had slept there for at least one night before the survey.

In this study, a total of 15,683 women of the reproductive age (15–49), 12,688 men aged 15–59 years in 16,650 households were interviewed in 2016 EDHS. Of these households, 1,682

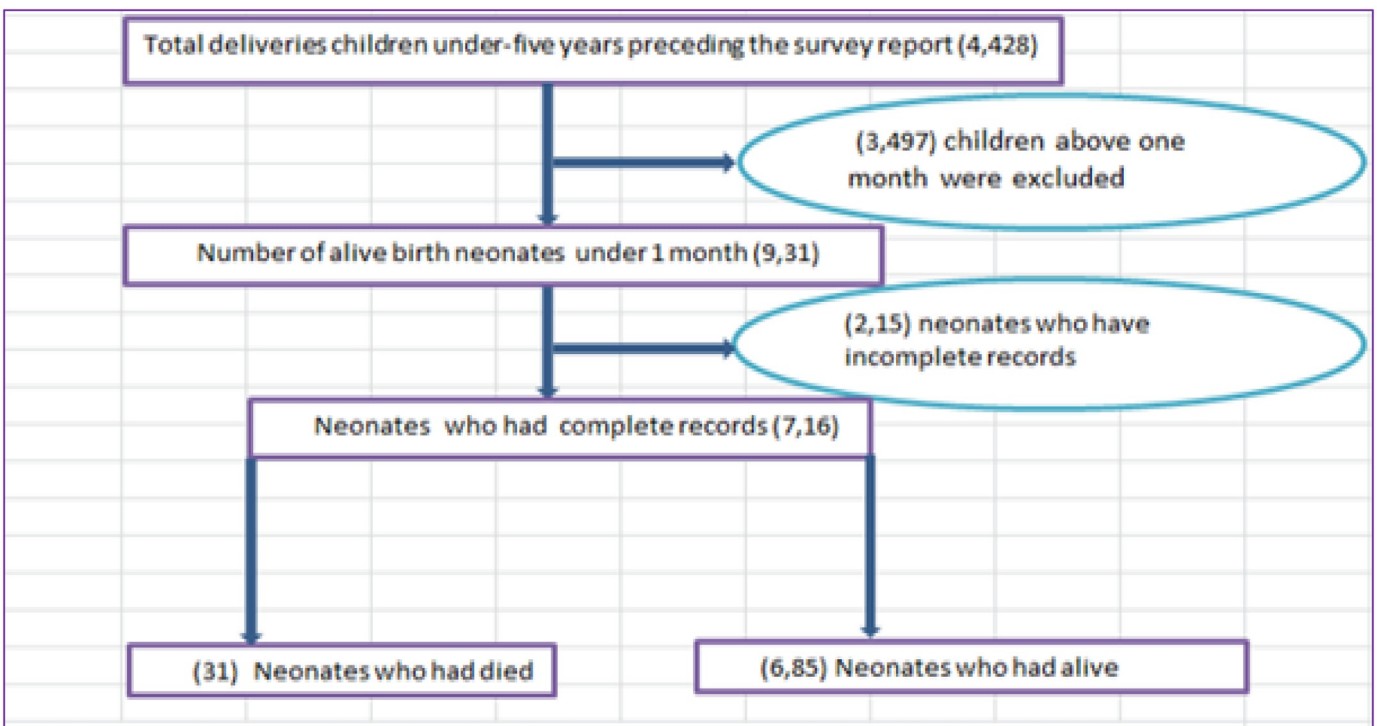

**Fig 1. Diagrammatic presentation of sample selection among neonates included in this study, 2016.**

were in the regional state of Tigray. The Tigray regional state recorded 4,428 live births in the five years preceding this 2016 EDHS survey.

Next, neonates having completed records within the preceding five years were located using the sample selection technique depicted in Fig 1 [18]. Finally, samples containing a total of 716 newborns were included, for whom comprehensive information encompassing all risk factors considered were accessible.

## Variables of the study

**Response variable.** In the EDHS 2016 survey, mothers were asked to report any live baby deaths and miscarriages. In this study, the response variable was the neonatal mortality rate.

Thus, the outcome variable for the $i^{th}$ neonate is dichotomous, represented by a random variable $Y_i$ that takes the value "1" with probability of success (neonatal death) and the value "0" with probability of failure (no neonatal death), such that

$$Y_i = \begin{cases} 1, \text{ if } i^{th} \text{ neonates had experienced neonatal mortality.} \\ 0, \text{ if } i^{th} \text{ neonates had not exprienced neonatal mortality.} \end{cases}$$

**Independent variable.** The sociodemographic characters of the mother and the newborn, which are listed in Table 1, served as the study's independent variables.

## Data processing and analysis

A statistical analysis was carried out using the statistical program Statistical Package for Social Science (SPSS), version 26. Using descriptive statistics, such as frequency distributions and

**Table 1. Operational definition and category of independent variables used in the study.**

| Variables | Description and categories |
|---|---|
| Mother's age | Mother's age (35 and above, 20–34, 19 and below) |
| Birth type | Birth type neonate (singletons, multiple) |
| Sex of neonate | Sex of neonate (female, male) |
| ANC follow-up | ANC follow-up (no, yes) |
| Toilet facility | Toilet facility using of mother's (with facility, no facility) |
| Mother's education level | Mother's education level (no education, primary and secondary, tertiary) |
| Residence | Residence place of neonates (rural, urban) |
| Drinking water source | Drinking water source of mother's (protected, piped, unprotected) |
| Place of delivery | Place of delivery of neonate (health facility, home) |

percentages, the study looked at every variable. Tables and hands were also used in the data-presenting process. One of the most widely used regression models in logistic analysis is the binary logistic regression model. This method was used to estimate the event probability for a categorical response variable with two outcomes.

The log odds model is linearly related to the predictors, such that

$$Y_i = \mathrm{Log}\left[\frac{\pi_i}{1-\pi_i}\right] = b_0 + b_1 x_{i1} + b_2 x_{i2} \ldots + b_p x_{ip.} \tag{1}$$

This was done in order to determine and assess the influence of each predictor on neonate death. Where, $Y_i = Log\left[\frac{\pi_i}{1-\pi_i}\right]$ = is the value of the unobserved outcome variable for the i[th] case of experiencing the death of neonate's.

$\pi_i$ = is the probability i[th] case of experiencing death of neonate's.

$1-\pi_i$ = is the probability of surviving in the previous five year of live birth neonate's.

$b_0$ = is log odds of the intercept.

$x_i$ = is the j[th] predictor for the i[th] birth.

$b_j$ = is the j[th] coefficient.

p = is the number of predictors.

The effect of covariate variables on newborn mortality was assessed using a multivariable binary logistic regression model. Calculations were used to determine the adjusted odds ratios (AOR) and their 95% confidence intervals (CI). Using the Hosmer and Lemeshow test and the Likelihood Ratio Test (LRT), we first examine the general goodness of fit. Accordingly, the LRT test provided a chi-square value of 76.012 with 14 degrees of freedom and a p-value of 0.000, which would imply a good fit for the model. Accordingly, the Hosmer-Lemeshow test revealed that the model more accurately predicted the observed data (chi-square value = 12.145 with 8 degrees of freedom and p-values = 0.145.

## Ethical consideration

In order to use the 2016 EDHS dataset from the DHS program for the current study, a formal consent letter was discovered. The dataset was also given IRB approval for public use without individual or household identity. As a result, the participants' identities were kept secret. Furthermore, this dataset was exclusively used for the current study in accordance with the DHS program discipline, rules, and regulations. However, access to the study's data set requires legal registration at https://dhsprogram.com/data/available-datasets.cfm and the creation of a persuasive letter outlining the project's descriptive goal for the DHS program.

## Results

### Descriptive statistics

A total of 716 neonates were included in the study to investigate the predictor factors for newborn death throughout the five years prior to the survey. According to Table 2, out of the 716 live births in the study sample, 31 (4.3% (95% CI: 3.06, 6.10) had been reported as neonatal deaths. According to the study, newborn mortality rates were 43 per 1000 live births for the five years prior to the survey. Table 2 displays the sociodemographic and economic characteristics of mothers and newborns for the studies identified risk variables of neonatal death.

About one-sixth of mothers' were in the age group 20–34 (60.9%), and nearly one-twentieth of mothers' were below or equal to 19 years, and 34.2% were in other age groups of 35 and above, respectively.

Regarding the birth type of neonates, more than half (97.3%) were singleton neonates. On the other hand, when the sex of the neonate was concerned, half (50.1%) of the participants reported being female neonates. Additionally, research shows that almost 20% of respondents were drinking unprotected well water and that more than half (52.0%) lacked toilet facilities.

The highest proportion of neonatal death rates occurs in mothers of age groups 35 and above (8.6%) and 19 and below (2.9%), respectively. As mothers get older and younger, they experience higher neonatal death rates. Neonatal death rates were highest (6.4%) among mothers with no schooling at all. Moreover, mothers of multiple birth types experienced a greater rate (31.6%) of neonatal deaths compared to mothers of singletons (3.6%) (Table 2).

**Table 2. Descriptive characteristics of the respondents in the Tigray regional state, Ethiopia, 2016 (n = 716).**

| Variables | Categories | Neonate survival status | | Total frequency (%) |
|---|---|---|---|---|
| | | Alive (%) | Died (%) | |
| Mother's age | 19 and below | 34 (97.1%) | 1 (2.9%) | 35 (4.9%) |
| | 20–34 | 427 (97.9%) | 9 (2.1%) | 436 (60.9%) |
| | 35 and above | 224 (91.4%) | 21 (8.6%) | 245 (34.2%) |
| Birth type | Multiple | 13 (68.4%) | 6 (31.6%) | 19 (2.7%) |
| | Singletons | 672 (96.4%) | 25 (3.6%) | 697 (97.3%) |
| Sex of neonate | Male | 333 (93.3%) | 24 (6.7%) | 357 (49.9%) |
| | Female | 352 (98.1%) | 7 (2.0%) | 359 (50.1%) |
| ANC follow-up | Yes | 574 (96.5%) | 21 (3.5%) | 595 (83.1%) |
| | No | 111 (91.7%) | 10 (8.3%) | 121 (16.9%) |
| Toilet facility | No facility | 355 (95.4%) | 17 (4.6%) | 372 (52.0%) |
| | With facility | 330 (95.9%) | 14 (4.1%) | 344 (48.0%) |
| Mother's education level | Secondary and above | 372 (96.1%) | 15 (3.9%) | 387 (54.1%) |
| | Primary | 92 (98.9%) | 1 (1.1%) | 93 (13.0%) |
| | No education | 221 (93.6%) | 15 (6.4%) | 236 (33.0%) |
| Residence | Urban | 20 (90.9%) | 2 (9.1%) | 22 (3.1%) |
| | Rural | 665 (95.8%) | 29 (4.2%) | 694 (96.9%) |
| Drinking water source | Unprotected | 131 (97%) | 4 (3.0%) | 135 (18.9%) |
| | Piped | 170 (95%) | 9 (5.0%) | 179 (25%) |
| | Protected | 384 (95.5%) | 18 (4.5%) | 402 (56.1%) |
| Place of delivery | Health facility | 368 (92.9%) | 28 (7.1%) | 396 (55.3%) |
| | Home | 317 (99.1%) | 3 (0.94%) | 320 (44.7%) |
| Total number of neonates under one month | | 685 (95.7%) | 31 (4.3%) | 716 (100%) |

## Predictor factors of neonatal mortality among neonates in the Tigray regional state

**Binary logistic regression model analysis.** A binary logistic regression model analysis was used to ascertain the effect of each predictor variable on newborn mortality.

## Predictor factors associated with neonatal mortality among neonates

Table 3 displays the results of an investigation of potential predictor factors linked to neonatal death using both a bivariable and a multivariable binary logistic regression model. From the bivariable analyses, birth type, sex of the neonate, mother's age, ANC follow-up, and place of delivery were significant predictors of neonatal mortality in neonates.

After adjusting for other predictor variables, the result showed that birth type, sex of the neonate, mother's age, place of delivery, and birth order were statistically significant predictors of neonatal death. However, ANC follow-up, toilet facility, mother's education level, residence, and drinking water source were not statistically significant.

According to the results from Table 3, the odds of neonatal mortality were higher among neonates that were multiple (AOR = 15.3, 95% CI: 3.54, 65.84) than those in singletons births.

**Table 3. The bi-variable and multivariable binary logistic regression analysis of predictor factors associated with neonatal mortality among neonates in the Tigray regional state, 2016 (n = 716).**

| Variables | Bi-variable binary logistic | Multivariable binary logistic |
|---|---|---|
| | COR (95% CI for COR) | AOR (95% CI for AOR) |
| **Mother's age (ref. = 35 and above)** | 1 | 1 |
| 19 and below | 0.314 (0.041, 2.41) | 0.263 (0.026, 2.692) |
| 20–34 | 0.225 (0.101, 0.499)* | 0.23 (0.087, 0.58)* |
| **Birth type (ref. = singletons)** | 1 | 1 |
| Multiple | 12.4 (4.4, 35.3)* | 15.3 (3.54, 65.84)* |
| **Sex of neonate (ref. = female)** | 1 | 1 |
| Male | 3.6 (1.54, 8.52)* | 3.75 (1.45, 9.75)* |
| ANC follow-up (ref. = no) | 1 | 1 |
| Yes | 0.41 (0.19, 0.890)* | 0.20 (0.025, 1.63) |
| Toilet facility (ref. = with facility) | 1 | 1 |
| No facility | 1.13 (0.55, 2.33) | 2.06 (0.613, 6.95) |
| Mother's education level (ref. = no education) | 1 | 1 |
| Secondary and above | 0.60 (0.29, 1.24) | 0.46 (0.125, 1.67) |
| Primary | 0.20 (0.021, 1.23) | 0.22 (0.024, 2.06) |
| Residence (ref. = rural) | 1 | 1 |
| Urban | 2.3 (0.51, 10.3) | 1.11 (0.085, 14.37) |
| Drinking water source (ref. = protected) | 1 | 1 |
| Unprotected | 0.65 (0.22, 1.96) | 0.54 (0.16, 1.81) |
| Piped | 1.13 (0.50, 2.57) | 0.35 (0.048, 2.59) |
| **Place of delivery (ref. = health facility)** | 1 | 1 |
| Home | 8.0 (2.42, 26.7)* | 7.4 (2.0, 27.6)* |
| **Birth order (ref. = 1st)** | 1 | 1 |
| 2–4 | 2.14 (0.83, 5.60) | 4.88 (1.52, 15.7)* |
| 5 and above | 1.17 (0.45, 3.03) | 1.10 (0.335, 3.32) |

* indicates significant predictors which have p-value <0.05, ANC: Antenatal Care,

**NB**: LRT = 76.012 (p < 0.000),

Hosmer-Leshow Test = 12.145 (p-value = 0.145)

Male neonates had a higher risk of mortality compared to females. The odds of neonatal mortality rates were 4 times higher among male neonates (AOR = 3.75, 95% CI: 1.45, 9.75) than females.

Concerning place of delivery, the odds of neonatal death rates were higher among neonates with no health facility (at home) (AOR = 7.4, 95% CI: 2.0, 27.6) compared to those with a health facility. Table 3 also shows that the odds of neonatal death rates were lower among neonates born to mothers whose ages were 20–34 (AOR = 0.23, 95% CI: 0.087–0.58) compared to those who were older than or equal to 35 years old. However, there was no significant difference in the predictors of neonatal mortality rate among mothers whose ages were 19 and below. Neonatal mortality was shown to be substantially correlated with birth order. The odds of neonatal death rates were about 5 times higher among neonates whose birth order was 2–4 (AOR = 4.88, 95% CI: 1.52, 15.7) than those of the 1st birth order (see Table 3).

## Discussion

Using the 2016 EDHS dataset, the current study empirically explored and determined the potential risk factors that were linked to neonatal mortality in the Tigray regional state. Thus, the prevalence of neonatal mortality in the Tigray regional state of Ethiopia was examined, along with its related risk variables, using the binary logistic regression model analysis. Neonatal mortality was 4.3% more common than average among neonates. This result was in line with the research conducted in Indonesia, 5.2% [19], Nigeria, 4.1% [10], Burkina Faso, 4.6% [20], the Somali region of Ethiopia, Jimma Zone, 3.55% [21], and Ethiopia, 3.67% [22]. However, the result of this study was lower than the study conducted at the University of Gondar Comprehensive Specialized Hospital in Northwest Ethiopia, 17.3% [8], Dilchora referral hospital in Dire Dawa, 11.4% [23], a sub-urban hospital in Cameroon, 15.7% [24], Wolaita Sodo Referral Hospital in southern Ethiopia, 17.3% [25], Amhara Regional State Referral Hospitals, Ethiopia, 18.6% [2], 16.7% [26], and 18.6% [27]. In contrast, it was higher than the study conducted in Debre Markos Referral Hospital in Northwest Ethiopia, 2.13% [6], China, 1.2% [27], rural areas of Eastern Ethiopia, 2.84% [28], and Sudan, 3.0% [29]. This discrepancy may be explained by variations in the study's design, settings, follow-up time, population sample size, and sociodemographics of its participants.

Multiple births were more likely than singletons to result in neonatal mortality. Male neonates had a higher neonatal death rate than female neonates, when all other factors were held equal. Similar earlier research conducted in the rural Umguza and Bubi districts also revealed a strong relationship between a neonate's sex and newborn mortality [2, 9, 30, 31]. As a result, women who have had multiple pregnancies should receive extra attention throughout ANC follow-up, prompt neonatal care following birth, and, to the greatest extent feasible, early breastfeeding initiation. Moreover, Genetic variations between male and female children may be the cause of this sex difference [31].

The place of delivery was discovered to be a major risk factor for neonatal mortality, meaning that newborns born at home had a larger risk of dying than newborns who received care in health institutions. Studies conducted in the past have found that newborns born in health facilities had greater access to delivery services and better healthcare services for them babies [8, 32].

Additionally, Table 3 showed that neonates born to women who were older had a noticeably lower risk of neonatal mortality. This demonstrates once again how older women were less likely to encounter neonatal mortality. Additionally, earlier studies have shown an inverse relationship between maternal age and neonatal mortality [2, 30]. Birth order was a key demographic predictor factor that was significantly linked to neonatal mortality. The neonatal

mortality was greater for newborns born in the second to fourth positions. This result is in line with the results of the research done at Arba Minch General Hospital in Ethiopia [33]. The rise toward childbearing among parents and the decreased use of maternity services seen in emerging nations are two potential factors.

Moreover, newborn mortality was unaffected by the availability of facility and ANC follow-up. These factors do, however, appear to have a major impact on infant mortality in certain research [34, 35].

Finally, since it is based on the national survey data set the investigation has the potential to give insight for program planners and policy-makers to design suitable intervention strategies both at regional and national levels.

## Limitation of the study

This study had limitations in that the 2016 EDHS report is frequently according to respondents' self-report due to this reason this might have the possibility of recall bias and also some significant predictor variables, such as the gestational mother's age, were left out of the study. Moreover, the study was done 07 years back so it is unlikely to reflect the latest status of neonatal mortality in the region.

## Conclusions and recommendations

The study recognized the sex of the neonate, birth order, mother's age, place of delivery, and birth type as potential risk factors for neonatal mortality. The prevalence of neonatal mortality indicated that the neonatal mortality rate in Tigray regional state was higher than the national average. To reduce neonatal mortality, targeted interventions should focus on high-risk groups, such as mothers delivering at home and those with multiple births. Additionally, they ought to make people aware of community-based longitudinal and survival studies, which can be used to identify additional, unmeasured risk factors for neonatal mortality.

## Acknowledgments

The author seriously acknowledges MEASURE DHS for giving way access to the EDHS dataset.

## Author Contributions

**Conceptualization:** Gebru Gebremeskel Gebrerufael, Brhane Gebrehiwot Welegebrial, Mehari Gebre Teklezgi.

**Data curation:** Gebru Gebremeskel Gebrerufael.

**Formal analysis:** Gebru Gebremeskel Gebrerufael, Brhane Gebrehiwot Welegebrial, Mehari Gebre Teklezgi.

**Investigation:** Gebru Gebremeskel Gebrerufael, Brhane Gebrehiwot Welegebrial.

**Methodology:** Gebru Gebremeskel Gebrerufael, Brhane Gebrehiwot Welegebrial, Mehari Gebre Teklezgi.

**Software:** Gebru Gebremeskel Gebrerufael, Brhane Gebrehiwot Welegebrial.

**Supervision:** Mehari Gebre Teklezgi.

**Validation:** Gebru Gebremeskel Gebrerufael.

**Visualization:** Gebru Gebremeskel Gebrerufael, Brhane Gebrehiwot Welegebrial, Mehari Gebre Teklezgi.

**Writing – original draft:** Gebru Gebremeskel Gebrerufael, Brhane Gebrehiwot Welegebrial, Mehari Gebre Teklezgi.

**Writing – review & editing:** Gebru Gebremeskel Gebrerufael, Brhane Gebrehiwot Welegebrial, Mehari Gebre Teklezgi.

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
