## [Decision Letter · Decision Letter 0]

16 Sep 2024

PONE-D-24-30270Predictors of neonatal mortality among neonates in Tigray regional state, Ethiopia: a retrospective cross-sectional studyPLOS ONE

Dear Dr. Gebrerufael,

Thank you for submitting your manuscript to PLOS ONE. After careful consideration, we feel that it has merit but does not fully meet PLOS ONE’s publication criteria as it currently stands. Therefore, we invite you to submit a revised version of the manuscript that addresses the points raised during the review process.

We look forward to receiving your revised manuscript.

Kind regards,

Kahsu Gebrekidan, Ph.D.

Academic Editor

PLOS ONE

Journal Requirements:

Reviewers' comments:

Reviewer's Responses to Questions

**Comments to the Author**

1. Is the manuscript technically sound, and do the data support the conclusions?

Reviewer #1: No

Reviewer #2: Partly

Reviewer #3: Partly

2. Has the statistical analysis been performed appropriately and rigorously? 

Reviewer #1: Yes

Reviewer #2: N/A

Reviewer #3: Yes

3. Have the authors made all data underlying the findings in their manuscript fully available?

Reviewer #1: No

Reviewer #2: Yes

Reviewer #3: Yes

4. Is the manuscript presented in an intelligible fashion and written in standard English?

Reviewer #1: Yes

Reviewer #2: No

Reviewer #3: No

5. Review Comments to the Author

Reviewer #1: This manuscript titled "Predictors of Neonatal Mortality among Neonates in Tigray Regional State, Ethiopia: A Retrospective Cross-sectional Study" focuses on neonatal mortality, a critical concern, especially in low-resource settings such as Ethiopia. The use of secondary data from the 2016 Ethiopia Demographic and Health Survey provides a solid foundation for identifying risk factors, and the application of multivariable logistic regression for data analysis is appropriate for the study objectives. The study contributes valuable insights to the existing body of knowledge regarding neonatal mortality at the regional level in Tigray and its predictors.

Constructive feedback:

1. L71: “As a result, this report shows that it has continued to perform at high level.” This statement is incongruent with the rest of the paragraph and does not specify what the performance is about or whose performance it is referring to.

2. Study design: In this study, the authors used secondary data from the DHS, there is no temporarily neither in data collection nor in participant enrolment. Even the DHS data collection was at one point in time. Therefore, the term “retrospective” does not fit into the study design. It is a cross-sectional study by analysis of secondary data.

3. Figure 1: Do not use comas to separate figures of a number when it is not up to a thousand.

4. L123: Do not begin a sentence with figures

5. L124: “have all been questioned since the 2016 EDHS.” This is confusing. Has there been additional questioning of these families after the DHS? What was the purpose f the additional questioning?

6. L126 “Preceding the research.” Please be consistent with differentiating between your study and the DHS. Specify whether the word “research” here refers to the DHS or your study.

7. L127 The word “locate” insinuates that contact was made with participants. Consider changing to “identified”

8. Your methodology does not clearly describe inclusion and exclusion criteria for the DHS nor those for your study selection. You do mention including cases with complete records but you have not specified which records (variables) you were interested in from the DHS.

9. L143 The description of neonatal death is incorrect. Aparently only deaths in the first week of life and pregnancy losses after 4 weeks of gestation were considered. This is not consistent with the usual estimation of neonatal mortality rates which includes deaths in the first 28 days of life and excludes pregnancy losses.

10. L199 The higest proportion of neonatal deaths, not death rates

11. L202 “in a similar neonatal….” This phrase is unclear

12. L202: “moms of multiple birth types” Do you mean moms who had multiple babies from the same pregnancy or moms who had several types of birth?

13. L255 change Cameron to Cameroon

14. In the discussion section, add a paragraph on how the study findings can be useful to policy makers in the strife to reduce neonatal mortality rates, with possible examples of strategies that have been applied in other settings where similar predictors were found.

15. Add a section on limitations of the study.

16. L279 You earlier mention that Ethiopia achieved MDG with reduction of U-5 mortality, but neonatal mortality rate increased (L70). Here you are making a contradicting statement.

17. L97 – 104 Write out your objective and specific objectives in a paragraph format, not bullet points.

18. L211-217 This description of methods of analysis should be moved to the methodology section of the manuscript.

19. The authors report adjusted odd ratios but do not tell what was adjusted for.

Reviewer #2: Abstract section

The sentence the predictors aren’t well recognized could be rephrased to be more formal and precise. A clearer version could be: the predictors are not well understood.

Neonates born to the mother’s age (20–34 years) – this phrase is slightly confusing. It would be clearer to say: neonates born to mothers aged 20–34 years

“Tigray regional state was experiencing a higher neonatal mortality rate than the national level” could be rephrased for clarity: The neonatal mortality rate in Tigray regional state was higher than the national average.

The conclusion section could be more impactful by briefly emphasizing the implications of the findings. For example: “To reduce neonatal mortality, targeted interventions should focus on high-risk groups, such as mothers delivering at home and those with multiple births.”

The sentences “male neonates were more likely to die” is somewhat abrupt and could be softened for tone. Consider: “Male neonates had a higher risk of mortality compared to females.”

Introduction section:

The sentence “The probability that a neonate would die within the first 28 days of life, or the first four weeks of life, is known as the neonatal mortality rate (NMR)” is a bit lengthy and could be streamlined for clarity: “Neonatal mortality rate (NMR) refers to the probability of a neonate dying within the first 28 days of life.”

Repetitiveness:

There are some repetitive statements about neonatal mortality, particularly in the first few paragraphs. For instance, the information about the percentage of child deaths due to neonatal mortality is mentioned multiple times. You could combine these points for better readability.

Missing Gap Explanation:

While the introduction explains the high neonatal mortality rate in Tigray and Ethiopia, it could better emphasize the specific research gap that this study aims to fill. For example, you could more explicitly state: “Despite various studies on neonatal mortality in Ethiopia, research focused specifically on regional variations, especially in the Tigray region, is limited.” This would highlight the study's novelty.

Transition to Objectives:

The transition from the background information to the study objectives is abrupt. Adding a sentence that ties the context to the specific aims of the study would improve the flow. For instance: “Understanding these risk factors in the context of Tigray can provide targeted interventions and inform public health strategies.”

Minor Stylistic Adjustments:

In the sentence “the neonatal mortality rate (NMR) accounts for about 44% of deaths in children under five”, it would be clearer to say: “Neonatal mortality accounts for approximately 44% of deaths in children under five.”

Method section:

Sample Size and Population

The methods section mentions that 716 neonates were included in the analysis. However, the justification for why this sample size was selected and whether it is statistically sufficient for identifying risk factors is not clearly addressed. Adding a brief explanation of sample size determination or its adequacy would enhance clarity.

Lack of Explanation on Missing Data:

It’s unclear how missing data were handled. Since secondary datasets often have incomplete data, it would be important to state if any neonates were excluded due to missing information or how missing data were managed (e.g., imputation methods, exclusion).

Ethics Considerations:

Although it is mentioned that ethical approval was not required due to the use of publicly available data, this section could be expanded to clarify how data confidentiality and anonymity were maintained.

Data Analysis Section Could Be Expanded:

While the use of binary logistic regression is mentioned, there is little detail about the selection of covariates or how multicollinearity was checked. It would strengthen the methodology to explain how variables were chosen for the model and whether tests for multicollinearity or interactions between variables were conducted.

Flow of Methods:

• The flow of the methods section can be improved by reorganizing the sections more logically. For instance, after explaining the study design, data source, and sample size, the section should discuss variables and their definitions, followed by the statistical analysis plan. The current flow jumps between these topics, which can be confusing.

Suggestion: Consider this sequence for the methods:

1. Study design and data source.

2. Sample size and population.

3. Definitions of dependent and independent variables.

4. Data analysis (descriptive and regression models).

5. Ethical considerations.

Results section:

Overuse of Numbers:

In the narrative portion of the results, there is an overemphasis on reporting numbers and percentages without sufficient explanation. While numbers are important, the results section could benefit from more interpretation and context around these findings.

Interpretation of Tables:

While tables are provided, there is minimal interpretation of the data in the narrative. The text mainly repeats the numbers found in the tables. More effort could be made to explain the patterns or trends observed in the data.

Goodness-of-Fit Test:

The results briefly mention that the Likelihood Ratio Test (LRT) and Hosmer-Lemeshow test were used to assess model fit, but there is little interpretation of what these results mean. This could confuse readers unfamiliar with these tests.

Subtle Presentation of Non-Significant Findings:

Some findings, such as ANC follow-up and toilet facility, are reported as non-significant predictors, but these could be better highlighted. Readers might overlook these aspects, which are important in understanding what did not contribute to neonatal mortality.

Discussion section:

Limited Discussion of Non-Significant Findings:

The discussion focuses heavily on significant predictors but does not adequately address the factors that were not significant (e.g., antenatal care (ANC) follow-up, mother's education, and toilet facilities). Discussing these non-significant results could provide important insights, such as why some expected predictors did not show a relationship with neonatal mortality in this region.

Inconsistent Depth in Comparisons with Other Studies:

While some comparisons with previous studies are well-done, others are underdeveloped. For example, the finding related to birth order could be discussed in more detail, as other studies might have identified different trends.

Lack of Explanation for Unexpected Findings:

Some findings, like the reduced neonatal mortality risk for mothers aged 20–34, could be explored more thoroughly. The discussion doesn’t sufficiently explain why younger mothers (under 20) did not show the expected increased risk of neonatal mortality.

Overemphasis on General Findings:

While the discussion covers the main findings well, it sometimes reiterates the results without adding much interpretation. For example, the finding about male neonates being more likely to die is mentioned without deeper exploration of why this might be the case beyond biological vulnerability.

Recommendations Could Be More Specific:

While the study provides general recommendations, they could be more targeted. For instance, rather than simply suggesting improvements in health services, the discussion could highlight specific interventions like community education on safe delivery practices, promotion of facility-based births, or specialized care for multiple births.

Need for More Regional Comparisons:

Although the discussion compares the findings to national and international data, it could be enhanced by more detailed regional comparisons within Ethiopia. This would help highlight whether the situation in Tigray is unique or reflective of broader trends in the country.

Limitations Section Could Be Expanded:

The limitations of the study are mentioned but could be more detailed. For instance, the reliance on secondary data from EDHS means there may be important variables that were not captured or measured accurately, such as quality of health services or cultural practices.

Reviewer #3: General comments

The research is scientifically sound and will significantly contribute to the scientific body of knowledge, particularly in the region of Tigray. However, a few areas need to be improved and proofread. I tracked and changed the document; you can find the comments, suggestions, and changes.

Questions and suggestions

1. There is 2019 EDHS mini data, and some research has already been published using the data. Why did you not use the recently updated data?

2. Background page 4, line 89: A few studies have been conducted in the Amhara, Sidama, and other regions.

3. Method and materials page 5, line 108: Did you conduct a survey? I think you accessed secondary data from the EDHS database.

4. Result section page 10, line 219: Did you calculate relative risk? I thought you used the ODDS ratio to predict factors influencing neonatal death.

5. Result section page 11, line 231: Where do you calculate 75%? I tried calculating the percentage, but it did not generate this number. Do you consistently use the ODDS ratio to interpret your findings?

6. Result section page 11, line 238: Is this a relative risk?

7. Result section page 11, line 240: Where do you calculate 88%? I tried calculating the percentage, but it did not generate this number. Do you consistently use the ODDS ratio to interpret your findings?

8. Discussion section, page 12, line 260: How will this variable possibly cause the discrepancy?

9. Discussion section page 12, lines 272-274: The possible explanation of the discrepancy of the findings.

10. Conclusion and recommendation section, page 13, lines 287-294: The recommendations are not adequate, especially for identified factors such as home delivery.

6. PLOS authors have the option to publish the peer review history of their article (what does this mean?). If published, this will include your full peer review and any attached files.

Reviewer #1: No

Reviewer #2: No

Reviewer #3: No

---

## [Author Response · Author response to Decision Letter 0]

23 Oct 2024

To: PLoS ONE Chief Editor,

 22-Oct-2024 

Dear Dr. Editor-in-Chief,

We are writing to submit our manuscript entitled, “Predictors of neonatal mortality among neonates in Tigray regional state, Ethiopia: a cross-sectional study” for consideration as PLoS ONE research article.

The author confirms that this work is original and hasn’t been published elsewhere, nor is it currently under consideration for publication elsewhere. Neonatal mortality remains one of the major public health issues in the world. Sub-Saharan Africa is one of the major public health problems affected, with the highest prevalence rate, particularly in Ethiopia.

Although many researchers have worked on the identification of potential risk factors associated with under-five children mortality, tremendous efforts have also been made by the Ethiopian government. However, neonatal mortality is still devastating and remains a major public health problem among neonates in Ethiopia, especially like in Tigray regional state. Most of these studies done in Ethiopia are concentrated on the national level. Such studies miss an important point for policymakers and designers, the results at the national level result may not show the exact situation at regional levels.

To our knowledge, there are limited studies that look at the national level of the prevalence rate and the potential predictor factors associated with neonatal mortality. To alleviate these problems, the author conducted a holistic community-based cross-sectional study of the present 2016 EDHS report. Therefore, our findings will allow your readers to understand the predictor factors involved in identifying the potential risk factors and to design policies at the regional level in order to control and reduce the major risk factors of neonatal mortality in Tigray Regional State, northern Ethiopia. 

We believe that the findings presented in our paper will appeal to public health service providers and policymakers who subscribe to the PLoS ONE. The author has no conflicts of interest to disclose. The author has substantially contributed to conducting the underlying research and drafting this manuscript. If you require any additional information regarding our manuscript, please don’t hastate to contact us directly via the resources below.

Thank you for your time and consideration of this manuscript.

Sincerely,

Gebru Gebremeskel Gebrerufael (Corresponding author)

E-mail address: gebrugebremeskel2015@gmail.com

Department of Statistics, College of natural science, Adigrat University, Adigrat, Ethiopia

---

## [Decision Letter · Decision Letter 1]

26 Nov 2024

Predictors of neonatal mortality among neonates in Tigray regional state, Ethiopia: a cross-sectional study

PONE-D-24-30270R1

Dear Mr. Gebru,

We’re pleased to inform you that your manuscript has been judged scientifically suitable for publication and will be formally accepted for publication once it meets all outstanding technical requirements.

Kind regards,

Kahsu Gebrekidan, Ph.D.

Academic Editor

PLOS ONE

Additional Editor Comments (optional):

Reviewers' comments:

Reviewer's Responses to Questions

**Comments to the Author**

1. If the authors have adequately addressed your comments raised in a previous round of review and you feel that this manuscript is now acceptable for publication, you may indicate that here to bypass the “Comments to the Author” section, enter your conflict of interest statement in the “Confidential to Editor” section, and submit your "Accept" recommendation.

Reviewer #1: All comments have been addressed

Reviewer #3: All comments have been addressed

2. Is the manuscript technically sound, and do the data support the conclusions?

Reviewer #1: Yes

Reviewer #3: Yes

3. Has the statistical analysis been performed appropriately and rigorously? 

Reviewer #1: Yes

Reviewer #3: Yes

4. Have the authors made all data underlying the findings in their manuscript fully available?

Reviewer #1: Yes

Reviewer #3: Yes

5. Is the manuscript presented in an intelligible fashion and written in standard English?

Reviewer #1: Yes

Reviewer #3: Yes

6. Review Comments to the Author

Reviewer #1: Congratulations for this important paper that shows a gap in reduction of neonatal deaths in the Tigray region compared to the rest of Ethiopia. You have identified vulnerable groups that could be specifically targeted by public health interventions.

Reviewer #3: The authors addressed all comments and suggestions. I believe the research will have a good scientific input, particularly in Ethiopia, specifically in Tigrai region.

7. PLOS authors have the option to publish the peer review history of their article (what does this mean?). If published, this will include your full peer review and any attached files.

Reviewer #1: No

Reviewer #3: No

---

## [Editor Report · Acceptance letter]

3 Dec 2024

PONE-D-24-30270R1 

PLOS ONE

Dear Dr. Gebrerufael, 

I'm pleased to inform you that your manuscript has been deemed suitable for publication in PLOS ONE. Congratulations! Your manuscript is now being handed over to our production team.

Kind regards, 

on behalf of

Dr. Kahsu Gebrekidan 

Academic Editor

PLOS ONE